# Brief communication: Threshold and probability. The conceptual difference between ID thresholds for landslide initiation and IDF curves

Francesco Marra[1], Eleonora Dallan[2], Marco Borga[2], Roberto Greco[3], and Thom Bogaard[4]

[1]Department of Geosciences, University of Padova, Padova, Italy
[2]Department of Land Environment Agriculture and Forestry, University of Padova, Legnaro, Italy
[3]Dipartimento di Ingegneria, Università degli Studi della Campania Luigi Vanvitelli, Aversa, Italy
[4]Department Water Management, Faculty of Civil Engineering and Geosciences, Delft University of Technology, Delft, the Netherlands

**Correspondence:** Francesco Marra (francesco.marra@unipd.it)

**Abstract.** Intensity-duration (ID) thresholds are used to identify rainfall conditions likely to initiate landslides. They consider the average rain intensity observed over the entire length (called duration) of user-defined wet periods that lead to the triggering. Intensity-duration-frequency (IDF) curves assign a probability to the intensity of precipitation observed over fixed-length temporal windows (also called durations). As the term duration refers to different concepts, ID thresholds and IDF curves cannot be compared directly, and should better not be plotted in one figure, and IDF curves should not be used to quantify the exceedance probability of ID thresholds.

## 1 Introduction

Regional early warning systems for shallow landslides and debris flows are often based on rainfall intensity-duration (ID) thresholds (Guzzetti et al., 2008; Segoni et al., 2018; Guzzetti et al., 2020). These thresholds are derived from landslide archives and rainfall observations with the aim of separating triggering and non-triggering events on the basis of their duration and average intensity (Leonarduzzi et al., 2017). Once these thresholds are defined, it is natural to ask the question "what is the probability of these conditions to occur?". Answering it is not trivial. IDF curves are the standard tool to calculate the annual exceedance probability of extreme rainfall intensities over a duration of interest (Kottegoda and Rosso, 2008). It follows that a seemingly natural way to quantify the annual exceedance probability of the rainfall causing landslides or debris flows is to calculate the return period of the intensity of ID thresholds using intensity-duration-frequency (IDF) curves. Indeed, IDF curves have been used to discuss the probability of ID thresholds, including works by the authors of this communication (e.g., Frattini et al., 2009; Destro et al., 2017; Bogaard and Greco, 2018). Here, we highlight an important conceptual difference between the *duration* used in ID thresholds and the *duration* used in IDF curves that has been overlooked by the landslide literature so far. We provide a real-world example and discuss some implications for the interpretation of shallow landslide and debris flow initiation.

## 2   Intensity-duration thresholds

Rainfall thresholds are rainfall conditions that, when exceeded, are likely to initiate landslides or debris flows (Guzzetti et al., 2008). Their actual occurrence is the result of the physical processes activating in slopes in response to precipitation (Bogaard and Greco, 2018). Nonetheless, as rainfall is the main trigger for these hazards, rainfall thresholds are among the most widely used tools to forecast the occurrence of landslides and debris flows on the regional scale. Following pioneering work by Caine (1980), the intensity $I$ and the length $D$ of the rain period that led to the triggering began to be used to determine the triggering conditions. Often, the total precipitation depth (for which the symbol $E$ is typically adopted) is used instead of the intensity, with no difference in the generality of our arguments since depth and intensity of any event are directly linked ($I = E/D$). Therefore, the *duration $D$* in this ID space is defined as the length of the wet period (that is, a user-defined event) that leads to the triggering, and the intensity $I$ refers to the average rain intensity observed during this period. Although landslides can be triggered by periods of high intensities that occur within rainfall events (D'Odorico et al., 2005; Moreno et al., 2025), the entire length of the events, or the length until the triggering time, if known, is used to build ID thresholds. It follows that the ID space is the space of the precipitation intensities $I$ observed over a $D$ that begins with the start of the precipitation event, however this is defined.

This approach requires an objective definition of rainfall events (Melillo et al., 2015), although subjective choices remain necessary in this context, as the user needs to define the criteria to separate rainfall events. In addition, rainfall records are often not available at hourly resolutions nor in close range of the landslide (Marra et al., 2016; Marra, 2019), which makes the events separation dependent also on these aspects. Different separation criteria and/or different resolution of the rainfall data would unavoidably lead to different definitions of the events, with different intensities and durations associated with the initiation of landslides. Due to the wide range of scales spanned by precipitation variability, ID thresholds often take the form of power laws (e.g., Caine, 1980, and the subsequent literature). Several approaches can be used to define these thresholds, including frequentist (Brunetti et al., 2010) and Bayesian methods (Berti et al., 2012), trained on triggering or non-triggering events only or triggering and non-triggering events together (Guzzetti et al., 2008; Peres and Cancelliere, 2021; Leonarduzzi et al., 2017).

## 3   Precipitation intensity-duration-frequency curves

Intensity-duration-frequency curves are a mathematical relationship among the rainfall intensity, the duration, and the annual frequency of exceedance (Koutsoyiannis et al., 1998). Intensity-duration-frequency curves are the most common tool used in hydrology and water resource engineering to quantify the annual exceedance probability (or frequency) of precipitation. Owing to the wide range of scales explored by precipitation variability, a fundamental parameter in the definition of precipitation probability is the temporal scale, that is defined as the time window of interest. For example this can be defined based on the typical response time of a hydrologic system, or on the typical time scales of the precipitation process of interest. This temporal window is usually called duration and the traditional symbol is $D$, but, to avoid misconceptions, in this communication we will use the symbol $W$. The *duration* of IDF curves, therefore, is a temporal running window of fixed length $W$. It follows that IDF curves provide the annual exceedance probability of an intensity $I$ over a temporal window of length $W$.

IDF curves are obtained by collecting the highest rainfall intensities observed any year over the time windows of interest. To do so, usually a running window of the desired length is moved across the timeseries and the largest values are extracted, for example the annual maxima or the exceedances of very high thresholds. Extreme value distributions are then used to describe these values, and intensities corresponding to an assigned cumulative probability are extracted for the required duration. A relationship is then estimated between the given intensities and the durations. Following simple scaling and multi-scaling arguments rooted in fractal theory, IDF curves are often described using scale invariance formulations, which, similarly to ID thresholds, take the form of power laws when observed across the temporal scale (Burlando and Rosso, 1996; Langousis and Veneziano, 2007).

## 4  Conceptual difference between ID thresholds and IDF curves

From the definition of ID thresholds and IDF curves it is clear that the time intervals over which the intensities of rainfall are examined in the two cases are different, although the term used to define them is the same. In fact, duration refers to the total length (or the time to triggering) $D$ of a user-defined rain event, on the one hand, and to a fixed-length temporal running window $W$, on the other. Once $W$ is chosen in IDF curves, the population of the corresponding intensities is defined *unconditionally* from the beginning of the event or from the landslide triggering moment. The duration $D$ in ID thresholds, instead, is defined *conditionally* to the beginning of the precipitation events. The population of ID pairs depends on user-defined choices concerning the identification of the events. Associating the ID pair of a landslide-triggering event to the IDF curve for the corresponding duration (and thus assigning this ID pair an exceedance probability based on the distribution of extreme events of that duration) disregards the conditions by which ID pairs are defined, that is, starting with the beginning of the event. The use of the same term for two conceptually different quantities, together with the common double-logarithmic transformation used in the plots, led to misunderstanding of these concepts and misinterpretation of the results.

For a given rainfall event, such as the ones of interest for ID thresholds, the intensity $I$ of IDF curves may refer to any window of length $W$ within the event, with $W = D$ being only one of the possible choices (Tsunetaka, 2021). Indeed, precipitation events are characterized by different return periods at different temporal windows (Bezak et al., 2016; Cache et al., 2025), and any temporal interval during the event could be the true, unknown, triggering interval $W^{\dagger}$. It follows that the length of the triggering rainfall $D$ used in ID thresholds does not necessarily coincide with the temporal window $W^{\dagger}$ that triggered the landslide. Even more crucially, the duration $D$ explored in ID thresholds cannot be objectively defined, because it drastically depends on how rain events are defined. Although several methods are available for objectively define triggering rain events (e.g. Melillo et al., 2015), they all necessarily rely on user-defined parameters.

The probability of observing an intensity $I$ over the duration $D$ of ID thresholds is a conditional probability, conditioned on the fact that the explored time interval ($D$) starts with the beginning of the precipitation event, however defined. Conversely, the probabilities given by IDF curves for a fixed window $W$ are unconditional. It is therefore erroneous to quantify the probability of a triggering event of intensity $I$ in the ID space of the ID thresholds using unconditional probabilities from the IW space of the IDF curves. Using the entire event length as $D$, for example when the exact time of occurrence of the landslide is unknown,

may often cause an underestimate of the severity of the triggering rainfall. As, by definition, events with lower severity are observed more frequently than events with higher severity, this underestimation may lead to false alarms when the information is used in real-time early warning systems. These false alarms add to the biases related to uncertain knowledge of the triggering moment and epistemic uncertainty on the triggering processes, and to the ones caused by systematic sampling of rainfall away from the location of the triggering landslide and by the use of coarse temporal resolution rainfall data (e.g, Marra et al., 2016; Marra, 2019). This same misconception has led to the conceptual error of thinking that long-duration rainfall (frequently up to 100 hours or more) triggers shallow landslides, while, sometimes, it is a high-intensity interval that occurs at favorable preconditions that causes the triggering (Bogaard and Greco, 2018).

## 5    Real-world example

The inconsistency highlighted in the previous section lies on a theoretical level, and holds for all the instances in which conditional and unconditional probabilities are confused. To make this theoretical issue more tangible, we provide here a real-world example. It should not be interpreted as an approach to solve the issue above, which cannot be solved. Further, the results we will present cannot be quantitatively exported to other cases. However, the qualitative implications in terms of direction of the biases are expected to hold.

We discuss the potential effects of the inconsistency highlighted above using data from 12 storms that triggered 133 debris flows in the eastern Italian Alps during 2005-2014. They constitute ∼40% of all the debris flows recorded in the area in this period (Nikolopoulos et al., 2014) and resulted from bulking and entrainment of unconsolidated channel-bed material. The triggering location of the debris flows in the database is provided with an uncertainty which is much smaller than the pixel size of the radar data (approximately one order of magnitude smaller, Marra et al., 2014). High-quality weather radar observations with resolutions of 1 km and 5 minutes are available for these events from Marra et al. (2014), but only information on the day of occurrence of the debris flows was available, while the exact time of occurrence was unknown. The dataset dates back over 10 years, but it remains among the few available datasets with quality-controlled continuous high-resolution rainfall estimates for many known landslides/debris-flows. We refer to this study for details on the weather radar correction procedures and on the quality of the derived time series. IDF curves for the area are available from Borga et al. (2005), which derived them using a scaling-invariant approach (see Kottegoda and Rosso, 2008).

For each debris flow, we extracted the precipitation time series observed by the radar over the triggering locations. This allows us to overcome the sampling limitations of the rain gauges (Marra et al., 2016). We identified the triggering events isolating them with at least 24 dry hours. Since information on the exact triggering time was not available, we used as $D$ the total length $D$ of the precipitation event and the average precipitation intensity $I$ observed over this time interval (duration concept of ID thresholds).

In a univariate framework, a possible choice of a return period $T^*$ representative of a rainfall event could be the maximum among the return periods $T_W$ associated with all possible temporal scales $W \leq D$: $T^* = \max(T_W)$. Here we will provide an example in which the triggering time interval is assumed to be the time interval during which the most severe intensity

was observed, meaning that we assume $W^\dagger = W^*$. It is important to note that this $W^*$ also depends on these user-defined parameters, because temporally-close heavy events may or may not be aggregated into one depending on these choices. For example two convective events occurring at a distance of 12 hours one from the other may be considered as one event when events are separated by dry periods of 24 hours and as two distinct events if separated by dry periods of 6 hours. Notably, Marra et al. (2020) showed that once independence is granted, this definition allows one to directly link the statistics of the

event maxima to the statistics of the annual maxima, thereby removing the ambiguity of rainfall event definition from IDF calculations.

We then examined the maximum intensity observed over a set of running windows of lengths $W$ of 5, 10, 15, 20, 30, 45 minutes and 1, 1.5, 2, 3, 4, 6, 9, 12, 18, 24, 36, 48 hours (duration concept of IDF curves). We identified the window over which the rain was the most severe (i.e., it had the lowest exceedance probability). To do so, we computed the return period

$T_W$ associated with the intensity of the rain observed on each of the windows $W$ by inverting equation (7) from Borga et al. (2005) and we identified the window $W^*$ in which the largest return period $T^*$ was observed.

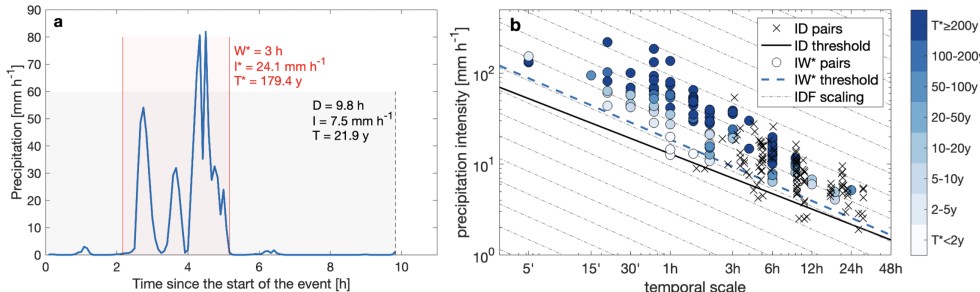

**Figure 1.** (a) Timeseries of a random triggering event. The vertical dashed line shows the end of the event, and black text reports the $I$, $D$, and $T$ values obtained considering the entire event as triggering. In red, the window over which the return period is the highest, with red text reporting $W^*$, the corresponding average intensity $I^*$ and return period $T^*$. (b) ID pairs of the triggering precipitation time series (black crosses) together with the ID threshold (black solid line). IW$^*$ pairs are shown as circles colored according to the return period. The dashed blue line shows the threshold one would obtain using the frequentist method on these IW$^*$ pairs. The average scaling invariance of the IDF curves for the region is shown in the background (dashed-dotted lines).

Figure 1 shows the ID pairs corresponding to the 133 debris flows as black crosses, together with the 5% ID threshold obtained using the frequentist method (Brunetti et al., 2010). The IW$^*$ pairs are shown as circles, and are colored according to the return period $T^*$, while the dashed blue line shows the threshold one would obtain using the frequentist method on the

IW$^*$ pairs. These latter threshold seem to better align with the regional scaling of extreme rainfall (dashed-dotted lines in the background), suggesting that the apparent difference in the power-law scaling of ID thresholds and IDF curves discussed by Bogaard and Greco (2018) can be attributed to methodological issues in the choice of rain duration, often made regardless of the physical processes responsible for debris flow or landslide occurrence. Here, the scaling of IW* pairs and ID pairs is

similar because our definition of $W^*$ follows what is done in IDF curves, but the intensities associated with the true $W^\dagger$ may

scale differently from the IDF curves, as they may be conceptually different.

In general, IW$^*$ pairs are associated with temporal scales $W^*$ that are always smaller than the duration $D$ of ID pairs (x-axis in Figure 1). In addition, by design, the corresponding intensities are systematically higher. This is a natural consequence of the temporal variability of precipitation (D'Odorico et al., 2005), and suggests that what is really important for triggering can be rain intensities over time scales much shorter than the total length of the identified rainfall event, as they are related with

the response time of the system in combination with the hydrological antecedent conditions. Indeed, for the 133 debris flows examined, the most severe intensities were observed for temporal windows $W$ between 30 minutes and 6 hours (Fig 2a). The severity on these time scales is about an order of magnitude higher than at other windows. Interestingly, these durations align with the critical durations for runoff generation in the catchments of the study area (Penna et al., 2017), where intense runoff is indeed the triggering mechanism of debris flows. These are also the time scales of convection, and encompass the scale of

individual convective cells, as well as the possible sequence of convective phenomena (Formetta et al., 2022). In fact, the vast majority of debris flows in the area (>90%) are associated with summer convective storms (Nikolopoulos et al., 2015).

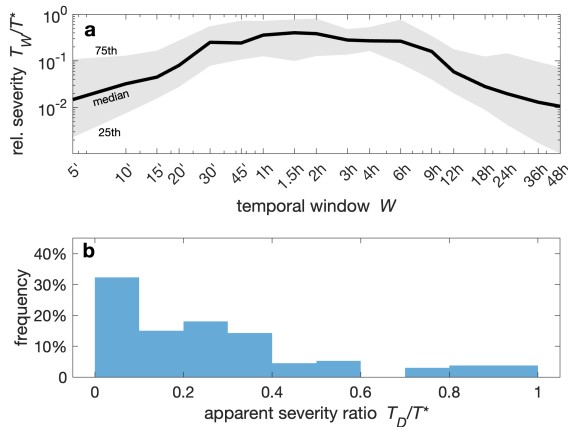

**Figure 2.** (a) Relative severity of the triggering precipitation for varying temporal window W. The relative severity is computed as the ratio between the return period at the time scale $W$, $T_W$, and the maximum return period $T^*$. The solid line and shaded areas represent the median and inter-quartile range across the 133 debris flows. (b) Apparent severity of the triggering precipitation obtained as the ratio between the return period of the precipitation over the entire storm length $D$, $T_D$, and the maximum return period $T^*$.

We then quantified how much the severity of the triggering precipitation is underestimated when using the entire length of the storm $D$ to calculate the return period $T_D$. Fig. 2b shows the ratio between the precipitation return period $T_D$ calculated for the intensity observed over the entire length of the storm $D$ and the maximum return period $T^*$ of the event. In ∼85%

of the cases, the return period estimated over the entire length of the storm is less than half than the maximum return period of the triggering precipitation. In about a third of the cases, it is underestimated by more than one order of magnitude. Once again, these differences are induced by the high temporal variability of the triggering precipitation. Indeed, Fig. 3 shows that

the typical decorrelation time of the debris-flow triggering precipitation, computed as the lag time at which the autocorrelation drops to $e^{-1}$, is on the order of 20-45 minutes (median 35 minutes), in line with what is expected of summer convection in the Alps.

The quantitative results presented for this study case strictly depend on the data we used (debris flow processes in the Alps, daily resolution on the occurrence of the debris flows, temporal resolution of the radar data, etc.), and on the methods and assumptions we took for designing the experiment (namely, that the triggering window corresponds to the window with the highest severity, i.e., $W^{\dagger} = W^{*}$). Nevertheless, they are expected to be qualitatively representative of the general case.

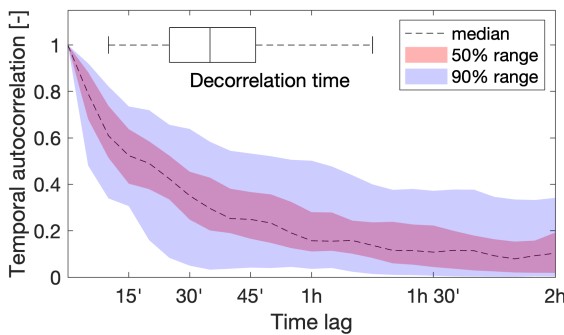

**Figure 3.** Temporal autocorrelation of the precipitation time series for the 133 debris-flow triggering events. Red and blue shaded areas show the 50% and 90% ranges across the 133 debris flow triggering events. The horizontal boxplot shows the distribution of the decorrelation times, calculated as the lag time at which the autocorrelation drops below $e^{-1}$.

## 6  Conclusions

We highlight the important conceptual difference between intensity-duration (ID) thresholds for landslides initiation and intensity-duration-frequency (IDF) curves used to calculate extreme rainfall probability. The term *duration* in the two refers to different concepts: the entire length of the triggering event in the case of ID thresholds, and a fixed-length running window in the case of IDF curves. Interestingly, the highlighted misconception may also explain the apparent difference in the power-law scaling of ID thresholds and IDF curves till 48 hours (e.g., Bogaard and Greco, 2018).

We provide a real-world example of such confusion for the case of debris-flow triggering thresholds in the eastern Italian Alps, showing that the most severe intensities were observed for temporal scales much shorter than the typical length of the triggering events. Estimating the probability of occurrence of these triggering conditions using IDF curves on ID pairs may cause an underestimation of the rainfall return period, especially when the exact time of triggering is not known and the entire event duration is used. This may lead to false alarms in early warning systems that operate in real time. In addition, representing ID thresholds over too long time scales may result in the wrong concept that shallow landslides and debris flows are triggered by long precipitation events while, in reality, given favorable preconditions, they may be triggered by heavy rain intensities

over relatively short time windows, the duration of which is related to the physical characteristics of the considered slope or catchment, as already pointed out over twenty years ago by Iida (1999) and D'Odorico et al. (2005).

In general, IDF curves should not be used to quantify the probability associated with ID thresholds (or ID pairs) and ID thresholds and IDF curves should not be plotted in the same graphs without clearly pointing out the conceptual difference between the two. So far, the highlighted inconsistency was overlooked by the community, leading to erroneous interpretations of probabilities. Alternative approaches to the definition of landslides triggering conditions which may be able to better capture intense rainfall periods during triggering events (e.g., Staley et al., 2017; Patton et al., 2023; Moreno et al., 2025) may be useful

in this sense.

*Code and data availability.*   Codes and data to reproduce the results and figures of this study are available at https://doi.org/10.5281/zenodo. 15845770 (Marra, 2025). The radar data were made available by the Autonomous Province of Bolzano. The parameters of the IDF model were taken from Borga et al. (2005).

*Author contributions.*   FM wrote the manuscript, analysed the data and prepared the figures. All authors contributed to the conceptualization

of the study and to revising and structuring the manuscript.

*Competing interests.*   At least one of the (co-)authors is a member of the editorial board of Natural Hazards and Earth System Sciences.

*Acknowledgements.*   We thank the editor and the anonymous reviewers for helping us improving the clarity of the paper. FM was partially supported by the "The Geosciences for Sustainable Development" project (Budget Ministero dell'Università e della Ricerca–Dipartimenti di Eccellenza 2023–2027 C93C23002690001). ED was supported by the RETURN Extended Partnership and received funding from the

European Union Next-GenerationEU (National Recovery and Resilience Plan – NRRP, Mission 4, Component 2, Investment 1.3 – D.D. 1243 2/8/2022, PE0000005).

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
