# Peer review of "Brief communication: Threshold and probability. The conceptual difference between ID thresholds for landslide initiation and IDF curves"

_EGUsphere, 2025_

## Author Comment (AC1)

**Referee #1**

We provide here our reply to the comments by Anonymous referee #1, together with the changes we plan to apply to the manuscript in response. The referee's comments are in black fonts and our reply in blue.

(1) This brief communication is very… brief! I mean, in the positive sense of the word. Indeed, it is a clear, concise manuscript that is perfectly written in fluent English - something very rare for a reviewer to find. I thank the authors for that! The paper gets straight to the point: landslide-triggering intensity-duration thresholds and precipitation intensity-duration-frequency curves cannot be confounded, compared, or plotted together. Neither one can be used to quantify the return time of the other.

Thank you for devoting your time to consider our work, this is much appreciated. We are also glad to hear that our conciseness was appreciated.

(2) Frankly, having worked on rainfall analysis and landslide prediction for years, the idea of mixing/comparing ID thresholds and IDF curves is something that never came to my mind. In the few cases I have seen in the extensive literature on these topics, it has always seemed very strange, not to say a downright methodological error. So, I can say that I certainly agree with the authors of this paper, although I do not think the article addresses a relevant scientific and/or technical question. I simply think that mixing ID thresholds and IDF curves is a misconception that does not even require discussion.

We agree the issue is indeed obvious when giving it a more profound thought. However, the misconception is to some extent also 'logical' with the used terminology and has been around implicit for quite some time in conference discussions and literature. Therefore, we think it can be useful to the community to clarify the differences and also indicate the consequences for landslide probabilistic analyses in climate change discussions.

(3) The authors list the differences between ID thresholds and IDF curves, focusing on the different durations (D and W) considered by the two tools, and then analysing the differences in terms of return time referring to these durations. In my opinion, they forgot the main and most important difference. That is: since their definition from pioneering works (Nel Caine and also previous pioneers), ID thresholds have been defined considering ID pairs that are somehow - arbitrarily or not, subjectively or not - linked to the initiation or re-activation of one or more landslides. On the other hand, IDF curves are defined considering IW (using the same terminology as the authors) pairs that are not linked to landslide/debris flow occurrence, referring only to rainfall itself. Indeed, the authors write "IDF are obtained by collecting the highest rainfall intensities observed any year over the time windows of

interest" (lines 45-46). Therefore, the two tools summarise or describe different variables (the ID pairs by which the thresholds are defined are different by definition from the IW pairs with pre-fixed durations of the IDF curves, having different characteristics consequently) and different processes (landslide or debris flow initiation and rainfall severity). This is, in my opinion, the main reason why the two tools must not be compared or mixed. I wouldn't have added anything else to this discussion

Reading this comment and some of the following, we realize that our message was not formulated clearly enough. While we agree that ID thresholds concern landslides/debris flow triggering or reactivation, we tend to disagree on the fact that this is the "main and most important" difference with IDF curves. The entire idea originates once ID pairs or thresholds are defined (in any way), and a perfectly natural question arises: "what is the probability of these conditions to occur?". The answer to this question is independent from the triggering of mass movements and only depends on the precipitation climatology of the area. What we would like to point out with our brief contribution is that the way that is often used to quantify this probability (by using IDF curves) hinges on a misconception. To make our point more clearly, we will include specific text in the introduction, as follows: "*[ …] events on the basis of their duration and average intensity (Leonarduzzi et al., 2017)*. **Once these thresholds are defined, it is natural to ask the question "what is the probability of these conditions to occur?"". Answering it is not trivial.** *IDF curves are [...]*".

However, the authors added more to the discussion, deserving attention. I list below some other comments on this paper.

(4) First, I don't understand the first part of the title "Threshold not probability". Actually, thresholds can be probabilistic. As a matter of fact, the Bayesian thresholds mentioned by the authors are probabilistic. Moreover, the frequentist thresholds also mentioned by the authors allow defining probabilistic diagrams to be used for early warning purposes. Therefore, I would remove this part of the title, which works only for deterministic, binary thresholds.

Thank you for this consideration. To our view, the fact that ID thresholds can be probabilistic does not make the title wrong. For that matter, IDF curves technically are thresholds: just like the definition of cumulative distribution function, that is non-exceedance probability of a given threshold. To address the reviewer concern, we modified the title as follows: *"Threshold **and** probability. The conceptual difference between ID thresholds for landslide initiation and IDF curves."*. Further, we will amend the text as described in our reply to the previous comment 3. This will be done early in the introduction, to make this aspect clearer right from the beginning: "*[ …] events on the basis of their duration and average intensity (Leonarduzzi et al., 2017)*. **Once these thresholds are defined, it is natural to ask the**

*question "what is the probability of these conditions to occur?'". Answering it is not trivial. IDF curves are [...]*".

(5) In several parts of the text, the authors write that quantifying the return period of a given intensity used to define ID thresholds using probabilities estimated from the IW space is erroneous and causes an underestimation of the severity of the triggering rainfall. I agree with the authors, totally. However, I'd suggest mentioning some works in which this erroneous approach was adopted, also because these are cited again in the last sentence of the paper ("Some results in the literature may thus be quantitatively inexact").

Thanks for this suggestion. References to some of these works are indeed reported in the manuscript, including the ones authored by ourselves (line 15 of the discussion paper). We intentionally did not provide a list of other instances to avoid the (unintended) implication that we want to challenge the main results of those studies, and we therefore removed the sentence mentioned by the referee.

(6) Moreover, I would add that the return period of a given ID thresholds should not be calculated at all. Indeed, rather than adopting dichotomous approaches (above/below threshold), using statistical and probabilistic approaches, as the two mentioned above, allows the probabilistic characterisation of the thresholds without introducing (erroneously) the concept of return time, which is also highly questionable for a variable not easily measurable as landslide or debris flow occurrence/triggering. In addition, as the authors certainly know, the concept of return time and how it changes in relation to non-stationarity is a topic of discussion in the scientific community.

We fully agree with the reviewer on these considerations. However, they seem to tackle a different, and much wider, problem: the one of how landslide triggering thresholds should be defined. Although most important, in this brief communication, we highlight a conceptual difference that hinges from theoretical arguments, with the aim of stimulating discussion within the community.

Moving to sections 2 and 3, the differences between ID thresholds and IDF curves are listed, focusing in particular on the different ways to define the duration of the ID/IW pairs.

(7) According to the authors' view, the durations D are user- (or arbitrary-) defined while the durations W are not. But, actually, W are also user- (or arbitrary-) defined using running windows of x minutes or hours: 5, 10, … 45 minutes or 1, 2, … 48 hours were also defined

by a user. Moreover, the authors didn't mention that IDF curves can be defined using the partial duration series approach as well, so introducing another point of discussion.

Thank you for this suggestion. We will rephrase this portion of the text to improve the clarity of our message. In IDF curves, the window W is set a priori based on some considerations (which we do not need to discuss here). At this point the probability of exceeding a given intensity over that window is estimated. What is relevant is that once W is chosen, the probability distribution of the extreme intensities observed over that time interval is well defined from a theoretical perspective (and it does not depend on the way we quantify it, annual maxima, partial duration series or other approaches). Conversely, the duration D in ID thresholds is defined *conditionally* to (a) the occurrence of a landslide, as correctly pointed out by the reviewer in comment 1, and also by (b) the identification of a triggering event. While statistically, it is possible to objectively define the population of ID pairs conditioned on (a), the population of ID pairs conditioned on (b) is not well defined because it depends on user-defined choices concerning the triggering event definition (different event definitions will lead to different populations and, hence, different statistics). This is because different definitions will cause the intensity to be averaged over a different time interval (or the depth to be accumulated).

We amended the text in section 4 to emphasize these aspects: "*In fact, duration refers to the total length of an **user-defined** rain event $D$, on the one hand, and to a fixed-length temporal running window $W$, on the other. **Once $W$ is chosen in IDF curves, the population of the corresponding intensities is theoretically well defined, and hence their probability distribution. The duration $D$ in ID thresholds, instead, is defined conditionally to (a) the occurrence of a landslide and to (b) the way triggering events are identified. While it is possible to objectively define the population of ID pairs conditioned on (a), the population of ID pairs conditioned on (b) depends on user-defined choices concerning the identification of the triggering events**. The use of the same term [...]*"

In addition, we amended the text to explicitly mention partial duration series: "*IDF are obtained by collecting the highest rainfall intensities observed any year over the time windows of interest. To do so, usually a running window of the desired length is moved across the timeseries and the largest values are extracted**, for example the annual maxima or the exceedances of very high thresholds**. Extreme value distributions are then used to describe these **values**, and intensities corresponding to an assigned cumulative probability are extracted for the required duration.*"

(8) In section 2 (lines 29-32) the authors write "rainfall records are often not available at hourly resolutions nor in close range of the landslide (Marra et al., 2016; Marra, 2019), which makes the events separation dependent also on these aspects.". Actually, this issue

affects the definitions of W too. Indeed, if only daily measurements are available in a given area, sub-daily values of W (e.g. the classical 1, 3, 6, 12, 24 hours) can't be defined, and the IDF curves cannot be drawn for sub-daily durations.

Thank you for commenting on this. Once again, here the referee focuses on the practical aspects while we are addressing a theoretical standpoint. In the case mentioned by the referee, it is not possible to *practically* calculate sub-daily values of *W* and draw the IDF curves, but the population and its statistics are well defined. Indeed, there are several approaches to estimate sub-daily IDF curves from daily observations, as well as sub-hourly IDF curves from hourly observations, via assumptions on the statistics of extremes across scales (e.g., Aronica & Freni, 2004; https://doi.org/10.1016/j.atmosres.2004.10.025).

(9) In section 4 (lines 62-63), the authors write "In a univariate framework, the return period T* of a rainfall event can reasonably be defined as the maximum among the return periods Tw associated with all possible temporal scales". I think that some examples should be provided to support this statement.

Thank you for this comment, which allows us to better specify some aspects of our reasoning that were not fully clarified. In theory, there is not one unique probability for a rainfall event, because this will depend on the examined scale (*W*, but one can think of areal scales as well, getting to the IDAF curves, with A standing for area). What we do here is provide a practical univariate solution, which, differently from the above points, cannot be backed theoretically. To make our study clearer, we reorganized the manuscript as follows.

(A) In section 4, we define W* as the true, unknown, triggering interval: *"Indeed, precipitation events are characterized by different return periods at different temporal windows**, and any temporal interval during the event could be the true, unknown triggering interval $W^*$. It follows that the length of the user-defined triggering rainfall $D$ does not necessarily coincide with the unknown temporal window $W^*$ that triggered the landslide (the equality $W^*= D$ only holds in very peculiar cases). *Even more crucially, [...]"* and "**Using the entire event length as $D$, for example when the exact time of occurrence of the landslide is unknown, almost always gives** $T_D \le T^*$, this erroneous approach **may often** cause a systematic underestimate of the severity of the triggering rainfall, leading to false alarms when the information is used in real-time early warning systems."*

(B) We move to the study case (section 5) our subjective choice in which W* is defined, only for the example case, as the one that maximizes the severity, rephrasing it as follows: *"**In a univariate framework, a possible choice of a return period $T^*$ representative of a rainfall event could be the maximum among the return periods $T_W$ associated with all possible temporal scales $W\le D$: $T^*=\mathrm{max}(T_W)$. Here we will provide an example in which the triggering time interval $W^*$ is assumed to be the**

*time interval during which the most severe intensity was observed. It is important to note that this $W^*$ also depends on these user-defined parameters, because temporally-close heavy events may or may not be aggregated into one depending on these choices. For example two convective events occurring at a distance of 12 hours one from the other may be considered as one event when events are separated by dry periods of 24 hours and as two distinct events if separated by dry periods of 6 hours. Notably, \citet{marra2020} showed that once independence is granted, this definition allows one to directly link the statistics of the event maxima to the statistics of the annual maxima, thereby removing the ambiguity of rainfall event definition from IDF calculations."*

Moving to section 5, I have some comments regarding the dataset used.

(10) First, it should be noted (and somewhere acknowledged by the authors) that the dataset is quite dated, having been collected over ten years ago.

Thank you for this observation. We added a sentence to section 5 to clarify this aspect: "**The dataset dates back over 10 years, but it remains among the few available datasets with quality-controlled continuous high-resolution rainfall estimates for many known landslides/debris-flows.** "

(11) Second, spatial and temporal information of the debris flow records is missing. In particular, authors should specify whether the time of occurrence is known for the debris flows included in the dataset used. This is extremely important information for a dataset to be used for the definition of rainfall thresholds. Moreover, it is relevant for another issue that I write further on in my comment.

Thank you for this comment. Indeed, information on the exact time of triggering was not available for all events, unfortunately. We amended section 5 to include it: "*High-quality weather radar observations with resolutions of 1 km and 5 minutes are available for these events from Marra et al. (2014)*, **but only information on the day of occurrence of the debris flows was available, while the exact time of occurrence was unknown.** *[...].* **Since information on the exact triggering time was not available, we used as $D$** *the total length $D$ of the precipitation event and the average precipitation intensity $I$ observed over this time interval (duration concept of ID thresholds).*" For what concerns the spatial information, we prefer to leave the reference to the studies in which the dataset was presented, as it is not important for our example and conclusions.

(12) Third, it is not described how the triggering precipitation events used to draw the thresholds were defined. This is also very relevant, given the comparison with IDF curves done in the paper.

Thank you for pointing out this critical aspect. We included the information as follows: "*We identified the triggering events isolating them with at least 24 dry hours.*"

(13) Further on in section 5, the authors describe the procedure used to calculate W* (lines 88-92). It should be acknowledged that the outcomes of this procedure are not related to debris flow triggering. Indeed, the fact that they have the highest return time among all IW pairs does not mean that they triggered debris flow. It would be useful to know when these IW* pairs occurred within the whole event duration, in order to establish whether they are relevant to the triggering of debris flows or not. If the IW* pairs occurred many hours (or days) before the occurrence of the debris flows, it cannot be said that they were certainly relevant to the initiation; at least, not more important than the entire event. This is the reason why knowing the exact time of occurrence of the debris flows is essential to prove that "what is really important for triggering are the rain intensities over time scales that can be much shorter than the total length of the identified rainfall event in combination with the hydrological antecedent conditions". In my opinion, selecting IW* pairs using the maximum return time as the only constraint is not sufficient to prove this hypothesis, and adds subjectivity in the process.

Thank you for this comment, which makes some very good points. The referee is correct: (i) the procedure is unrelated to the triggering, and (ii) the interval W* over which T is maximized could be unrelated to the triggering as well (as a matter of fact, in our study case it could even happen after the triggering, given that we are considering the entire event and we only have information about the day of the triggering. The objective of our data analyses is to show there is an inconsistency and demonstrate that an objective definition of W* leads to consistent representations of probability in the ID (or IW) space and we did not intend to provide a framework to assess these probabilities or to define thresholds.

These are important aspects that should be better explained in the manuscript, and we did extensive edits to the storyline. As detailed in our reply to comment 9 above (formal definition of W* in section 4 and practical definition for the example case).

Further, we rephrased section 2 to clarify some aspects and put more weight on the difference between unknown and known triggering times (note that knowing the triggering times helps in practice, but does not change our theoretical arguments) as follows: "*Following pioneering work by \citet{Caine1980}, the intensity $I$ and the length $D$ of the rain **period that led to the triggering** began to be used to determine the triggering conditions. **Often, the total precipitation depth ($E$) is used instead of the intensity, with no difference in the generality of our arguments since depth and intensity of any**"

*event are directly linked ($I=E/D$). Therefore, the \emph{duration} $D$ in this ID space is defined as the length of the wet period (that is, a user-defined event) that leads to the triggering, and the intensity $I$ refers to the average rain intensity observed during this period. Although landslides **can be** triggered by periods of high intensities that occur within rainfall events \citep{DOdorico2005, Moreno2025}, the entire length of the events**, or the length until the triggering time, if known,** is used to build ID thresholds*".

Last, to fully clarify that we are dealing with a hypothetical case, we rephrased the part of section 5 highlighted by the referee as follows: "*[...] and **suggests** that what is really important for triggering **can be** rain intensities over time scales that can be much shorter than the total length of the identified rainfall event**, as they are related with the response time of the system** in combination with the hydrological antecedent conditions. Indeed, for the 133 debris flows examined, the most severe intensities were observed for temporal windows $W$ between 30 minutes and 6 hours (Fig \ref{fig:f2}a). The severity on these time scales is about an order of magnitude higher than at other windows. **Interestingly, these durations align with the critical durations for runoff generation in the catchments of the study area \citep{Penna2017}, where intense runoff is indeed the triggering mechanism of debris flows**. These are **also** the time scales of convection …*"

(14) Then (lines 98-99), the authors write that "IW* pairs are associated with temporal scales W* that are always smaller than the duration D of ID pairs. In addition, by design, the corresponding intensities are systematically higher". This is tautological and led to what is written in lines 109-111 (i.e., the underestimation of the return times of the whole events compared to the IW* pairs). Again, having a lower return time does not imply that an ID pair is less severe in terms of landslide/debris flow triggering. This is another point to be added in the conceptual difference between ID thresholds and IDF curves.

Thank you for raising this consideration. The aim of our case study is to demonstrate with some numbers the impact of the misconception, and to introduce a possibly questionable but objective way to extract intensity-duration pairs (IW*) that belong to well defined populations. Indeed, using such pairs the mismatch between ID thresholds and IDF curve scaling shown in Bogaard & Greco, 2018 (reference in the manuscript) is solved. We believe that the restructuring detailed in our replies to comments 9 and 13 fully addresses this concern.

(15) Moreover, the authors assumed that ID thresholds are always defined considering D as the whole duration of the rainfall events. This is not always true. There are several examples in the literature in which sub-events are distinguished (automatically or not) within the entire rainfall events and used to define rainfall thresholds. This can be considered a

solution to the issues about durations being too long. I'd suggest mentioning it in the discussion.

Thank you for pointing this out, we amended the text in several places to address this issue. We'd like to underline that, while the practical issue is important, it has no implications toward our discussion.

(16) Before moving to the conclusions, two comments on Figs. 2 and 3. In Fig. 2, the (a) and (b) labels are missing. Fig. 3 and its description are not very clear; a better description a more discussion are needed.

Thank you for the suggestion, we added the labels to Figure 2. The caption of Figure 3 has been updated to: "Temporal autocorrelation of the precipitation time series for the 133 debris-flow triggering events. **Red and blue shaded areas show the 50\% and 90\% ranges across the 133 debris flow triggering events. The horizontal boxplot shows the distribution of the decorrelation times, calculated as the lag time at which the autocorrelation drops below $e^{-1}$.** "

(17) Going to the conclusions of the work, I totally agree that the calculation of return times of triggering conditions should be avoided, for several reasons including the ones described by the authors. However, the main motivation should be that it's better to use statistical/probabilistic approaches to define rainfall thresholds rather than calculating return times of the triggering conditions.

Our objective of this brief communication is to highlight a misconception in ID-IDF interpretation and consequently, we conclude that ID thresholds cannot be used to assess return times of triggering conditions. However, we do not aim to discuss how we can 'best' calculate landslide/debris flow activation probability.  In short, the motivation proposed by the referee to also discuss statistical/probabilistic approaches to define rainfall thresholds is beyond the scope of the brief communication, although we look forward to a scientific debate on that.

(18) Moreover, the underestimation of the return periods should be better evaluated considering the time of occurrence of the IW pairs and landslides/debris flows.

Thank you for this suggestion. We rephrased this part of the conclusions as follows: *"Estimating the probability of occurrence of these triggering conditions using IDF curves on ID pairs may cause an underestimation of the rainfall return period, **especially when the***

*exact time of triggering is not known and the entire event duration is used. This may lead to false alarms in early warning systems that operate in real time."*

(19) Overall, I think that the main message of the work is clear and shareable. However, I believe that the conclusions would need results based on an accurate dataset and improved methodology. In my opinion, more temporal details on the dataset are needed, in order to allow the most important methodological improvement needed in the work: that is, find the time of occurrence of the IW* pairs and their temporal distance from the debris flow occurrences. Only in this way will the conclusions be adequately justified by the data and results. So, my suggestion is that the work needs major revisions before being reconsidered for publication. The revised version of the paper should include an analysis of the temporal instants of the IW* pairs, so as to say with certainty that they can be considered the cause for debris-flow-triggering. This may be done using information from the proposed dataset (if any) or using other datasets. Moreover, I'd kindly suggest taking into consideration all my comments regarding theoretical and methodological aspects of the work.

Thanks again for the time devoted to our work and for the useful suggestions. We believe the amendments we proposed to the manuscript allowed us to clarify several important points and address the referee's concerns.

---

## Author Comment (AC2)

**Referee #2**

We provide here our reply to the comments by Anonymous referee #2, together with the changes we plan to apply to the manuscript in response. The referee's comments are in black fonts and our reply in blue.

Thank you for the opportunity to review Marra et al., "Brief communication: Threshold not probability. The conceptual difference between ID thresholds for landslide initiation and IDF curves." (egusphere-2025-3378). In this contribution, the authors explore the conceptual difference between rainfall intensity-duration (ID) thresholds for landslide initiation, which are conventionally fit to ID pairs that consider average intensity over entire landslide-triggering rainfall events with the intention of identifying conditions under which landslides are more likely, and intensity-duration-frequency (IDF) curves, which are fit to ID pairs that consider annual maximum average intensities for windows of defined duration with the intention of estimating annual exceedance probabilities. The authors argue that, because the definition of duration is different, these two curves are not comparable and IDF curves should not be used to estimate the exceedance probability of landslide-triggering rainfall. They use an example dataset of debris flows from the eastern Italian Alps to compare the implications of using the conventional approach based on the entire event duration to define I-D thresholds and an alternative approach that selects the duration with the maximum return period during an event. They show that return periods are much higher for the window with the maximum return period during the event than for the whole event. They also show that the slope of an ID threshold that uses the alternative approach better matches the slope of the regional IDF curves.

(1) Overall, this brief communication is well-written, thought-provoking, and has caused me to reconsider some results of my own research. It points out some important issues with ID thresholds that will be instructive for landslide researchers. In my view, the key contributions are (1) the clear explanation of how ID thresholds and IDF curves differ conceptually, (2) the insight that the return period of the average intensity over an entire event is the lowest possible return period and that much higher return periods may exist for shorter periods within an event, and (3) the recognition that if landslide triggering rainfall events are sampled with duration windows akin to the blocks used to determine IDF curves, the slope of ID threshold matches the regional IDF curves, at least for the case study presented. I believe that points (2) and (3) could be further emphasized in the text and should included in the abstract.

Thank you for the time devoted to our manuscript. We are glad to hear that our contribution was appreciated and considered worthy of publication.

Concerning the inclusion of points 2 and 3 in the abstract, we believe they should not be part of the main message of our communication, because:

- Point 2 is not necessarily true. It is generally likely, but counterexamples can be found, e.g. a synthetic event with two strong hourly peaks (e.g, 100 year return period each) separated by a dry hour. It is likely that the 3-hour return period will be the largest
- Point 3 is correct but including it in the abstract would require substantial background information that cannot be included in an abstract e.g., what do we mean by 'duration windows akin to the blocks used to determine IDF curves' and what the slope of ID and IDF are. Point 3 is, however, addressed in section 5, where we state: *"These latter threshold nicely aligns with the regional scaling of extreme rainfall (dashed-dotted lines in the background), solving the apparent difference in the power-law scaling of ID thresholds and IDF curves discussed by \citet{Bogaard2018}.* **This result confirms that this apparent difference can be attributed to methodological issues in the choice of rain duration, often made regardless of the physical processes responsible for debris flow or landslide occurrence.** *In general, IW$^*$ pairs are associated with temporal scales $W^*$ that are always smaller than the duration $D$ of ID pairs".*

Because this piece is likely to serve as a primer on this topic for future researchers, there are some arguments that need a more nuanced explanation and it must be made clear which points are the author's opinions and which are supported by the evidence presented. Additional references are needed throughout. In particular:

(2) The authors make the arguments that "IDF curves should not be used to quantify the exceedance probability of ID thresholds" (Lines 5 – 6) and "it is therefore erroneous to quantify the return period Td of a given intensity I in the ID space using probabilities estimated from the IW space of IDF curves." (Lines 70 – 71) From my perspective, it is not wrong, if one has made the conventional choice that W=D, to look up what the exceedance probability of that ID pair is. The key issue lies in making the choice that W=D in the first place, as this choice obscures shorter periods of high intensities that may have much lower exceedance probabilities, as shown in the case study. This distinction needs to be made very clearly. The "should" in the first statement and the "erroneous" in the second are based on the opinion that it would be better to use W* to define the exceedance probability of the event than choosing W=D. While I tend to agree, this short paper does not present evidence that W* is a better predictor of triggering rainfall than W=D, so it needs to be clear that this is the authors' opinion.

We respectfully disagree with the referee on this point. The choice W=D will lead to the inconsistency. We will try to better detail why.

Once W is chosen, the probability distribution of the extreme intensities observed over that time interval is well defined from a theoretical perspective (and it does not depend on the way we quantify it, annual maxima, partial duration series or other approaches).

Conversely, the duration D in ID thresholds is defined *conditionally* to (a) the occurrence of a landslide, as correctly pointed out by the reviewer in comment 1, and also by (b) the identification of a triggering event. While statistically, it is possible to objectively define the population of ID pairs conditioned on (a), the population of ID pairs conditioned on (b) is not well defined because it depends on user-defined choices concerning the triggering event definition (different event definitions will lead to different populations and, hence, different statistics). This is because different definitions will cause the intensity to be averaged over a different time interval (or the depth to be accumulated).

One can choose W=D, but the statistics of W are the same no matter how events are defined, but the statistics of D depend on when the event starts. Let's see an example: we have one wet hour with 10 mm, then 12 dry hours and then 1 wet hour with 18 mm that leads to a landslide. If we separate our events with 24 dry hours, the two wet periods would constitute one event with D=14h and I=28mm/14h=2mm/h. If we separate the events with 6 dry hours, the two wet periods would be two distinct events, and the triggering ID pair would be D=1h and I=18mm/h. We can continue the reasoning with different separations without finding a solution.

We amended the text in section 4 to emphasize this critical aspect: "*In fact, duration refers to the total length of an **user-defined** rain event $D$, on the one hand, and to a fixed-length temporal running window $W$, on the other. **Once $W$ is chosen in IDF curves, the population of the corresponding intensities is theoretically well defined, and hence their probability distribution. The duration $D$ in ID thresholds, instead, is defined conditionally to (a) the occurrence of a landslide and to (b) the way triggering events are identified. While it is possible to objectively define the population of ID pairs conditioned on (a), the population of ID pairs conditioned on (b) depends on user-defined choices concerning the identification of the triggering events**. The use of the same term [...]*"

Further, we added new text in section 5 as follows: "***For example two convective events occurring at a distance of 12 hours one from the other may be considered as one event when events are separated by dry periods of 24 hours and as two distinct events if separated by dry periods of 6 hours. Notably, \citet{marra2020} showed that once independence is granted, this definition allows one to directly link the statistics of the event maxima to the statistics of the annual maxima, thereby removing the ambiguity of rainfall event definition from IDF calculations.***"

It follows that our "should" and "erroneous" mentioned by the referee are not based on an opinion, rather on the difference between probability concepts highlighted above.

(3) The analysis shows that an ID threshold fit to IW* pairs better matches the slope of the regional IDF curves than a conventional threshold and the authors argue that this solves

"the apparent difference in the power-law scaling of ID thresholds and IDF curves discussed by Bogaard and Greco (2018)." This is an interesting result, my interpretation of which is that when time series of debris flow triggering rain are sampled with W*, the method is similar enough to using block maxima that the distribution of extreme rainfall is similarly represented. That would suggest that the difference between ID and IDF slopes can be attributed to methodological differences in how rainfall time series are sampled rather than any physical processes. If the authors agree that this is the case, I recommend making this point explicitly to avoid any further confusion. But then I have to wonder – what about filling-storing-draining?

Thank you for this thought-provoking point. Replacing D with W* leads to events that belong to the same type of population, so that they align with the slope of the IDF curve. This is an argument from a purely statistical viewpoint and as such, correct. This particular result comes from the fact that the maxima over a duration W of independent events share the tail statistics of the annual maxima (see Marra & al 2020, cited in the discussion paper).

Clearly, this brief communication focuses on the incorrect link between the statistics of rainfall time series versus the statistics of the amount of water needed to (re-)activate a landslide. However, the hydrological cause is – in a physical sense – responsible for the variation in meteorological triggers for initiating landslides. Filling-storing-draining is a conceptual framework for the physics of delayed response, variation in threshold volumes of water and the timing of landslides. Phrased differently: The probabilistic mismatch between IDF and ID threshold has nothing to do with the physical interpretation of the landslide triggering thresholds, but at the same time it depends on the physical processes behind the ID (or depth-duration ED) pairs. The point here is that the exceedance probability of an ID pair, even assuming it can be unambiguously calculated, is not the probability of a landslide or debris flow triggered by a rainfall of duration D. This latter, in fact, depends also on the physical response of the slope (as we already pointed out in our reply to a previous comment, it is indeed a conditional probability). As the focus of this brief communication is on the statistical consequences, we then made a few changes/additions to the text to avoid misunderstanding about the role of the physical processes.

We made this aspect clearer right from the beginning of the introduction, as follows: *"These thresholds are derived from landslide archives and rainfall observations with the aim of separating triggering and non-triggering events on the basis of their duration and average intensity \citep{Leonarduzzi2017}. **Once these thresholds are defined, it is natural to ask the question ``what is the probability of these conditions to occur?''. Answering it is not trivial.** IDF curves are the standard tool to calculate the annual exceedance probability of extreme rainfall intensities over a duration of interest \citep{Kottegoda2008}."*

We amended the text to explicitly mention the probabilistic mismatch as follows: ***"In a univariate framework, a possible choice of a return period $T^*$ representative of a rainfall event could be the maximum among the return periods $T_W$ associated***

*with all possible temporal scales $W \le D$: $T^* = \mathrm{max}(T_W)$. Here we will provide an example in which the triggering time interval $W^*$ is assumed to be the time interval during which the most severe intensity was observed. It is important to note that this $W^*$ also depends on these user-defined parameters, because temporally-close heavy events may or may not be aggregated into one depending on these choices. For example two convective events occurring at a distance of 12 hours one from the other may be considered as one event when events are separated by dry periods of 24 hours and as two distinct events if separated by dry periods of 6 hours. Notably, \citet{marra2020} showed that once independence is granted, this definition allows one to directly link the statistics of the event maxima to the statistics of the annual maxima, thereby removing the ambiguity of rainfall event definition from IDF calculations.”*

Further, following the referee's suggestion, we included a comment on the slope, as follows: *“These latter threshold nicely aligns with the regional scaling of extreme rainfall (dashed-dotted lines in the background), solving the apparent difference in the power-law scaling of ID thresholds and IDF curves discussed by \citet{Bogaard2018}. **This result confirms that this apparent difference can be attributed to methodological issues in the choice of rain duration, often made regardless of the physical processes responsible for debris flow or landslide occurrence.**”*

(4) The authors note that corresponding intensities for W* are systematically higher than W=D, which they argue "Implies that what is really important for triggering are rain intensities over time scales that can be much shorter than the total length of the identified rainfall event in combination with the hydrological antecedent conditions" (Lines 99 – 102). I do not understand how the first point implies the second. There is a logical gap here that needs to be addressed.

Following the recommendations from referee #1, we softened this sentence to fully clarify that we are dealing with a hypothetical case: *"[...] and **suggests** that what is really important for triggering **can be** rain intensities over time scales that can be much shorter than the total length of the identified rainfall event**, as they are related with the response time of the system** in combination with the hydrological antecedent conditions. Indeed, for the 133 debris flows examined, the most severe intensities were observed for temporal windows $W$ between 30 minutes and 6 hours (Fig \ref{fig:f2}a). The severity on these time scales is about an order of magnitude higher than at other windows. **Interestingly, these durations align with the critical durations for runoff generation in the catchments of the study area \citep{Penna2017}, where intense runoff is indeed the triggering mechanism of debris flows**. These are **also** the time scales of convection …"*

I have some additional suggestions that I believe could make the manuscript more instructive, particularly for readers who are less familiar with ID, IDF, or both:

(5) In Figure 1, I suggest labeling the IDF scaling lines with return periods to make it more clear what these refer to.

Unfortunately this cannot be done because the 133 debris flows examined occurred in different places, where the statistics of extremes differ.

(6) I also suggest adding a panel to this figure that shows a time series of a debris flow triggering event with windows that show W* and W=D and the average intensities and their return periods over each of these windows. This will help readers to better understand the difference between the ID pairs and IW* pairs.

Thank you for the suggestion, we included a new panel in figure 1 as recommended.

(7) As an outlook in the conclusions, the authors may want to consider mentioning the variety of alternative approaches to determining thresholds or estimating continuous probabilities that are better able to capture intense periods in landslide triggering time series than averaging over the entire event. For example, both (Staley et al., 2017) and (Patton et al., 2023) compared models trained with accumulations over different windows to select a model that best separated triggering from non-triggering events for post-fire debris flows in the western United States and shallow landslides in Alaska. The (Moreno et al., 2025) study that was already cited is a nice example of how we can move away from the need to bin time series entirely.

Thank you for this interesting suggestion. We added a sentence to the conclusion in this regard: *"Alternative approaches to the definition of landslides triggering conditions which may be able to better capture intense rainfall periods during triggering events \citep[e.g.,][]{Staley2017,Patton2023,Moreno2025} may be useful in this sense."*

Line by line comments:

Line 9 – suggest citing (Guzzetti et al., 2020)

Thank you, we added the reference.

Line 58 – this statement needs a reference

The references to this were mentioned in the introduction, including references to works we authored (line 15 of the discussion paper). We prefer not to provide a list of other instances to avoid the feeling we want to challenge the results of other researchers. It is likely that in many cases only minor parts of the publications may be affected by this issue, and we trust that the reader will have enough information to understand which parts, should there be any, of a paper of interest could be less reliable.

Line 63 – this statement needs a reference and possibly more context. Is this choice a convention in meteorology or is this an argument that the authors are making here?

Thank you for raising this good point. This is a subjective argument of ours. In theory, there is not one unique probability for a rainfall event, because this will depend on the examined scale (*W*, but one can think of areal scales as well, getting to the IDAF curves, with A standing for area). What we do here is provide a practical univariate solution, which, differently from the above points, cannot be backed theoretically. To make our study clearer, we reorganized the manuscript as follows.

(A) In section 4, we define W* as the true, unknown, triggering interval: *"Indeed, precipitation events are characterized by different return periods at different temporal windows**, and any temporal interval during the event could be the true, unknown triggering interval $W^*$. It follows that the length of the user-defined triggering rainfall $D$ does not necessarily coincide with the unknown temporal window $W^*$ that triggered the landslide (the equality $W^*= D$ only holds in very peculiar cases).** Even more crucially, [...]"* and "**Using the entire event length as $D$, for example when the exact time of occurrence of the landslide is unknown, almost always gives** $T\_D \le T^*$, this erroneous approach **may often** cause a systematic underestimate of the severity of the triggering rainfall, leading to false alarms when the information is used in real-time early warning systems."

(B) We move to the study case (section 5) our subjective choice in which W* is defined, only for the example case, as the one that maximizes the severity, rephrasing it as follows: *"**In a univariate framework, a possible choice of a return period $T^*$ representative of a rainfall event could be the maximum among the return periods $T\_W$ associated with all possible temporal scales $W\le D$: $T^*=\mathrm{max}(T\_W)$. Here we will provide an example in which the triggering time interval $W^*$ is assumed to be the time interval during which the most severe intensity was observed. It is important to note that this $W^*$ also depends on these user-defined parameters, because temporally-close heavy events may or may not be aggregated into one depending on these choices. For example two convective events occurring at a distance of 12 hours one from the other may be considered as one event when events are separated by dry periods of 24 hours and as two distinct events if separated by dry periods of 6**

*hours. Notably, \citet{marra2020} showed that once independence is granted, this definition allows one to directly link the statistics of the event maxima to the statistics of the annual maxima, thereby removing the ambiguity of rainfall event definition from IDF calculations."*

Line 72 – Please add ~one sentence clarifying how this leads to false alarms.

We rephrased the sentence as follows: "**As, by definition, events with lower severity are observed more frequently than events with higher severity, this underestimation may lead** to false alarms when the information is used in real-time early warning systems"

Line 74 – this statement needs a reference

The sentence will be removed.

Line 81 – How did you select these 12 storms as opposed to considering all debris flow triggering storms? Please clarify.

Preparing quality controlled and corrected weather radar data using physically-based corrections is a time consuming activity. These events constitute the database prepared over the course of a PhD on radar hydrology (from the first author of this manuscript). We preferred events that triggered several debris flows, for which weather radar visibility in the area of interest was good, radar data was complete and with high quality. As mentioned in the discussion paper, these storms triggered 40% of the debris flows in the region during that period, and can therefore be considered a very good sample.

Line 87 – Please add one sentence detailing how you defined the events (e.g. length of dry period). As you noted earlier, the derived ID pairs are sensitive to these choices.

Thank you for pointing out this critical aspect. We included the information as follows: "*We identified the triggering events isolating them with at least 24 dry hours.*"

Line 94, Figure 1 – Please add an estimate of statistical uncertainty to these thresholds.

While uncertainty is fundamental in practical applications, here we are trying to communicate a theoretical point. Adding the uncertainty would not add useful information for our message.

Line 127 – surely, Moreno et al., 2025 aren't the first to point this out. Earlier reference?

We removed this reference and included the one recommended in the next comment.

Line 129 - (Iida, 1999) also noted this

Thanks, the reference was added.

References

Guzzetti, F., Gariano, S. L., Peruccacci, S., Brunetti, M. T., Marchesini, I., Rossi, M., and Melillo, M.: Geographical landslide early warning systems, Earth-Science Reviews, 200, 102973, https://doi.org/10.1016/j.earscirev.2019.102973, 2020.

Iida, T.: A stochastic hydro-geomorphological model for shallow landsliding due to rainstorm, CATENA, 34, 293–313, https://doi.org/10.1016/S0341-8162(98)00093-9, 1999.

Moreno, M., Lombardo, L., Steger, S., De Vugt, L., Zieher, T., Crespi, A., Marra, F., Van Westen, C., and Opitz, T.: Functional Regression for Space-Time Prediction of Precipitation-Induced Shallow Landslides in South Tyrol, Italy, JGR Earth Surface, 130, e2024JF008219, https://doi.org/10.1029/2024JF008219, 2025.

Patton, A. I., Luna, L. V., Roering, J. J., Jacobs, A., Korup, O., and Mirus, B. B.: Landslide initiation thresholds in data-sparse regions: application to landslide early warning criteria in Sitka, Alaska, USA, Natural Hazards and Earth System Sciences, 23, 3261–3284, https://doi.org/10.5194/nhess-23-3261-2023, 2023.

Staley, D. M., Negri, J. A., Kean, J. W., Laber, J. L., Tillery, A. C., and Youberg, A. M.: Prediction of spatially explicit rainfall intensity–duration thresholds for post-fire debris-flow generation in the western United States, Geomorphology, 278, 149–162, https://doi.org/10.1016/j.geomorph.2016.10.019, 2017.

---

## Author Response (AR2)

We provide here our reply to the comments from the Editor and the referees, together with the changes we plan to apply to the manuscript in response. The comments are in black fonts and our reply in blue.

**Editor**

Dear                                                                                                                      Authors,
Thank you very much for responding to the reviewers' comments in detail. Following your responses, some other aspects are of concern to the reviewers. I would appreciate it if you could do the necessary minor changes suggested by Reviewer 2. If you could quantify the influence of the uncertainty as indicated by Review 1, that would be greatly appreciated. Following your responses, I will resend the manuscript to the reviewers and ask for another round                                                of                                            recommendations.
Kind                                                                                                                     regards
Ugur                                                                                                                      Öztürk
NHESS Topic Editor

We would like to thank the editor for the prompt handling of our manuscript - much appreciated - and for the suggestions.

We are glad to see the positive evaluations from the referees. Despite this, after reading their comments, we realized that our main message was still somehow under-appreciated. We briefly clarify it here before getting into the point-by-point reply.

Our message concerns a theoretical issue that was mostly overlooked: the confusion between *conditional probabilities* and *marginal* (or *unconditional*) *probabilities*. We then provide a practical example, with the goal of making this theoretical issue more tangible. Most of the referee's attention is attracted by this last example, but they fail to focus on the real message - the theoretical one.

With this in mind, we rephrased several parts of the manuscript to better guide the attention of the readers on this aspect - please refer to the response to the reviewers.

We clearly state now that:

"*From the definition of ID thresholds and IDF curves it is clear that the time intervals over which the intensities of rainfall are examined in the two cases are different, although the term used to define them is the same. In fact, duration refers to the total length **(or the time to triggering) $D$** of a user-defined rain event, on the one hand, and to a fixed-length temporal running window $W$, on the other. Once $W$ is chosen in IDF curves, the population of the corresponding intensities is **defined \emph{unconditionally} from the beginning of the event or from the landslide triggering moment**. The duration $D$ in ID thresholds, instead, is defined \emph{conditionally} to **the beginning of the precipitation events**. The population of ID pairs depends on user-defined choices concerning the identification of the **events**. **Associating the ID pair of a landslide-triggering event to the IDF curve for the corresponding duration (and thus assigning this ID pair an exceedance probability**

*based on the distribution of extreme events of that duration) disregards the conditions by which ID pairs are defined, that is, starting with the beginning of the event."*

And that:

"*The probability of observing an intensity $I$ over the duration $D$ of ID thresholds is a conditional probability, and is conditioned on the fact that the explored time interval ($D$) starts with the beginning of the precipitation event, however it is defined. Conversely, the probabilities given by IDF curves for a fixed window $W$ are unconditional. It is therefore erroneous to quantify the probability of a given intensity $I$ in the ID space of the ID thresholds using unconditional probabilities from the IW space of the IDF curves."*

The sentences above are independent of the triggering processes and of the data source used in the example, instead they are general. Several examples that have nothing to do with landslides can be made. For instance, one would not assess the growth of a child by comparing the height of the child with the heights of the general population (unconditional). What one should do is to compare it with the children of the same age (conditional to the age). This clarifies the *conditional* versus *marginal* (or *unconditional*) issue. On top of that, in our case the age of the child (exact window of triggering) is not exactly known, but we only know that the age is "likely" lower or equal than, say, 10 years (where the "likely" refers to our epistemic uncertainty, and this limit refers to the time between the beginning of the precipitation event and either our best estimate of the triggering instant or the end of the precipitation event). This does not mean that conditioning the comparison to 10 years old children is appropriate, since we only know it is lower or equal than that.

Concerning the request from referee #1: the true, unknown, triggering window does not necessarily depend on the dataset used nor on the knowledge of the exact triggering instant. It is always true when the main assumption behind the use of ID thresholds (that is, the DF/landslides are not influenced by the precipitation that occurred before the event begins - i.e., before we start the clock that quantifies D) is met. This is because this assumption implies that the earliest possible start of triggering temporal window W^\dagger, while unknown, is the start of the storm, and the latest possible end of W^\dagger is the triggering instant, which is before the end of the event.

This is a general concept and does not depend on our data nor on its uncertainties. Running the requested analyses is therefore irrelevant to our message. Instead, it would increase the focus on a part of the example that is irrelevant to our main message, thereby distracting the attention of the readers from it.

To further improve clarity, we changed the notation for the true unknown triggering window (now $W^\dagger$) to avoid confusion with the assumed triggering window of Section 5 (W*). This came out of a suggestion from reviewer #2 that we believed got to the point of what confused the reviewers of the previous round.

We trust these clarifications explain why some of the aspects raised by the referees are not relevant to our message, and we hope the new version better clarifies these aspects to the readers as well.

**Referee #1**

I would like to thank the Authors for responding to my comments and clarifying most of my doubts. The manuscript certainly benefited from the review phase; some issues are now better clarified, and the main motivation of the work is more understandable.

Thank you for taking this additional time on our manuscript and providing your feedback.

I've still only one concern, regarding the way the Authors identified and compared the durations D and W. The authors state that "IW* pairs are associated with temporal scales W* that are always smaller than the duration D of ID pairs". Now we know that the triggering instants of the debris flows in the used dataset are not known. Thus, it is clear that the ID pairs, as calculated by the authors, could have included hours that were not related to the triggering: if a debris flow occurred in the middle of one day, the ID pair was defined including around 12 hours – and maybe a few mm of rainfall – which should have been discarded. As a consequence, the calculated ID pairs can be longer (and the intensity lower) than they should be. Therefore, the main reason for the fact that IW* were shorter than ID may lay in the dataset used (with only daily temporal information) and in the method used to calculate D. This has also implications in the discussion on the underestimation of the triggering precipitation (figure 2 and related text). This issue should be better acknowledged by the Authors (e.g., in lines 130-135), and – on the other hand – should suggest that the results of the real-world example cannot be easily generalized.

Thank you for this comment. The reasoning of the referee, however, is only partially correct. The reviewer concludes that "the main reason for the fact that IW* were shorter than ID may lay in the dataset used (with only daily temporal information) and in the method used to calculate D". This is incorrect. The fact that $W^* \leq D$ does not necessarily depend on the dataset used nor on the knowledge of the exact triggering instant. It is always true when the main assumption behind the use of ID thresholds (that is, the DF/landslides are not influenced by the precipitation that occurred before the event begins - i.e., before we start the clock that quantifies $D$) is met. This is because this assumption implies that the earliest possible start of triggering temporal window W*, while unknown, is the start of the storm, and the latest possible end of W* is the triggering instant, which is before the end of the event.

This being said, we agree on the fact that the results of a real-world example cannot be quantitatively generalized, although the qualitative aspect (direction of the bias) is general. We amended the text in this direction: *"The quantitative results presented for this study case strictly depend on the data we used (debris flow processes in the Alps, daily resolution on the occurrence of the debris flows, temporal resolution of the radar data, etc.), and on the methods and assumptions we took for designing the experiment (namely, that the triggering window corresponds to the window with the highest severity, i.e., $W^\dagger=W^*$). Nevertheless, they are expected to be qualitatively representative of the general case."*

My question is: is the underestimation of the triggering precipitation due to a real difference between IW* and ID pairs, or it is due to the coarse temporal resolution of the dataset (which affects only the calculation of ID)? Perhaps the Authors could try to quantify the influence of the uncertainty in the debris flows triggering instants on the calculation of the ID pairs, before comparing them with the IW* pairs. A way to address this issue can be found in some published works. A few years ago, Peres et al. (2018) [https://doi.org/10.5194/nhess-18-633-2018] proposed a quantitative analysis of the impacts of uncertain knowledge of landslide initiation instants on the assessment of rainfall thresholds, using a synthetic dataset. The authors found that uncertainties in the landslide triggering instants may lead to underestimation of the thresholds and consequently more false positives. More recently, Mondini et al. (2025) [https://doi.org/10.1016/j.scitotenv.2025.179453] characterized the temporal uncertainty of some records included in a landslide catalog before using them to build a prediction tool for rainfall-induced landslides.

I acknowledge that this would need some additional work (perhaps too much for a brief communication?), but this issue came out only after the first round of review, during which it was clear that the triggering instants of the debris flows were not known. The best option would be to carry out these analyses to remove any doubt. If the Editor considers this to be too much work for a brief communication, I believe it is at least necessary to discuss the uncertainties related to the data in greater detail

As mentioned in our reply to the comment above, the fact that $W^\dagger ≤ D$ (where $W^\dagger$ is the true, unknown, triggering window) does not necessarily depend on the dataset used nor on the knowledge of the exact triggering instant. It is always true when the main assumption behind the use of ID thresholds (that is, the DF/landslides are not influenced by the precipitation that occurred before the event begins - i.e., before we start the clock that quantifies $D$) is met. This is because this assumption implies that the earliest possible start of triggering temporal window W^\dagger, while unknown, is the start of the storm, and the latest possible end of W^\dagger is the triggering instant, which is before the end of the event.

This is a general concept and does not depend on our data nor on its uncertainties. Running the requested analyses is therefore irrelevant to our message. Additionally, we are afraid it would distract the readers from the main idea behind this study, that is the confusion between *conditional* and *marginal* (or *unconditional*) *probabilities* applied to the case of ID thresholds and IDF curves.

To better deliver our message, we included several edits to the main text, as follows:

Section 2: ***"It follows that the ID space is the space of the precipitation intensities $I$ observed over a $D$ that begins with the start of the precipitation event, however this is defined."***

Section 3: "***It follows that IDF curves provide the annual exceedance probability of an intensity $I$ over a temporal window of length $W$.***"

Section 4: *"From the definition of ID thresholds and IDF curves it is clear that the time intervals over which the intensities of rainfall are examined in the two cases are different, although the term used to define them is the same. In fact, duration refers to the total length **(or the time***

*to triggering) $D$ of a user-defined rain event, on the one hand, and to a fixed-length temporal running window $W$, on the other. Once $W$ is chosen in IDF curves, the population of the corresponding intensities is **defined \emph{unconditionally} from the beginning of the event or from the landslide triggering moment**. The duration $D$ in ID thresholds, instead, is defined \emph{conditionally} to **the beginning of the precipitation events**. **The** population of ID pairs depends on user-defined choices concerning the identification of the **events**. **Associating the ID pair of a landslide-triggering event to the IDF curve for the corresponding duration (and thus assigning this ID pair an exceedance probability based on the distribution of extreme events of that duration) disregards the conditions by which ID pairs are defined, that is, starting with the beginning of the event.** [...] **The probability of observing an intensity $I$ over the duration $D$ of ID thresholds is a conditional probability, and is conditioned on the fact that the explored time interval ($D$) starts with the beginning of the precipitation event, however it is defined. Conversely, the probabilities given by IDF curves for a fixed window $W$ are unconditional.** It is therefore erroneous to quantify the **probability** of a given intensity $I$ in the ID space of the ID thresholds using **unconditional** probabilities from the IW space of the IDF curves"*

Section 5: *"**These quantitative results strictly depend on the data we used (debris flow processes in the Alps, daily resolution on the occurrence of the debris flows, temporal resolution of the radar data, etc.), on the methods and assumptions we took for designing the experiment (namely, the triggering window $W^*$ corresponds to the window with the highest severity). Nevertheless, they are expected to be qualitative representative of the general case.**"*

**Referee #2**

I have reviewed the manuscript entitled "Brief communication: Threshold and probability. The conceptual difference between ID thresholds for landslide initiation and IDF curves" as an additional reviewer for this round. This paper examines the conceptual differences between the ID thresholds and IDF curves in relation to debris-flow and landslide triggering. In Sections 1–4, the authors clearly and concisely describe the essential issues regarding rainfall data processing for ID thresholds and IDF curves. In Section 5, based on a well-constrained debris-flow dataset, the authors convincingly demonstrate how these processing differences can affect our understanding of rainfall conditions that trigger debris flows. The results are robust and highly meaningful, and will likely stimulate further analyses of rainfall-induced landslide and debris-flow initiation processes. Although the presented framework requires independent IDF curves, it is widely applicable across diverse environmental settings and effectively removes the dependence on local conditions and subjective interpretation. Overall, this paper represents a necessary and timely contribution to our community and could serve as a benchmark study. It is a promising and well-written manuscript that I believe will reach a wide audience. I have a few comments and suggestions that may help improve the clarity and overall flow of the paper. I hope my feedback will assist the authors in further refining this excellent                                                                                                                        work.
Sincerely,
Haruka Tsunetaka

Thank you for spending time and effort on our manuscript. We are glad to read such a positive evaluation. We did our best to address your concerns and suggestions, as detailed below.

In general, we realise that several of the raised points concern section 5, that is a simple example we propose to make the theoretical reasoning in section 4 a bit more tangible. It is important to clarify this to avoid the readers putting too much weight on the technical details of this section instead of our main message. For this reason, we now included a new paragraph at the beginning of section 5: *"The inconsistency highlighted in the previous section lies on a theoretical level, and holds for all the instances in which conditional and unconditional probabilities are confused. To make this theoretical issue more tangible, we provide here a real-world example. It should not be interpreted as an approach to solve the issue above, which cannot be solved. The results we will present cannot be quantitatively exported to other cases, but its qualitative implications in terms of direction of the biases are expected to hold."*

1.        Differences        between        debris        flows        and        landslides
My first concern relates to the differences between debris flows and landslides in the triggering mechanisms implicitly evaluated by ID thresholds. As described in L23-24, ID thresholds for landslide triggering commonly evaluate whether slope is activated "trigger" and "cause" prepared by rainfall input (Bogaad and Greco, 2018). However, some researchers argue that ID thresholds for debris-flow triggering reflect various processes, such as changes in sediment availability (e.g., Pastorello et al., 2018; Tsunetaka et al., 2021a), whether debris flows reach to the monitoring station (e.g., Bel et al., 2017), and changes in sediment composition (e.g., Guo et al., 2016; Tsunetaka et al., 2021b). These differences, which ID pairs may evaluate different initiation mechanisms between landslides and debris flows, should        be        well        considered        through        the        manuscript.
In my view, these differences relate to only interpretation of real world example (i.e., Section 5). Thus, my recommendation is that, either deleting or moving the related sentence (L23-24) and paragraph (L78-87) to Section 5, or providing a more precise explanation of the above differences. By doing so, the explanations provided in Sections 1–4 would more clearly highlight that they describe generalized issues concerning the ID–IDF relationship, which apply universally, regardless of whether the triggering process regarding debris flows or landslides.

Thank you for this consideration. The reviewer correctly evaluates that the reasoning about trigger and cause may only relate to the example in section 5. To better clarify this aspect, we amended a paragraph in section 4, that now reads: *"These false alarms add to the **biases related to uncertain knowledge of the triggering moment and epistemic uncertainty on the triggering processes and to the** ones caused by systematic sampling of rainfall away from the location of the triggering landslide and by the use of coarse temporal resolution rainfall data."*

Concerning the sentence in the introduction, we'd like to highlight that lines 23-24 do not mention the "trigger-cause" concept. Actually, they provide a caveat on the fact that the effects of precipitation are mediated by physical processes which then lead to the possibility of triggering. For this reason, we believe that the statement fits where it currently is (introduction).

2. Difference in the definition of W* between Sections 4 and 5
In the previous review round, both reviewers raised several concerns regarding the analysis presented in Section 5. In my view, these concerns mainly stemmed from the difference in the definition of W* between Sections 4 (L72: true, unknown triggering interval) and 5 (L105: time interval during which the most severe intensity was observed). In the revised manuscript, the authors have addressed this issue by clearly describing how W* is defined. However, I am still concerned that this difference may cause readers to misunderstand how the authors distinguish between theoretical facts and their interpretations throughout the paper. Indeed, it appears that the authors themselves may still be somewhat uncertain about this distinction (L139–140). It might be clearer to readers if, in Section 5, all results were consistently described in terms of W corresponding to max(Tw), with the subsequent discussion and interpretation developed under the assumption that such W is approximately equivalent to W*.

Thank you for this comment, we think it goes right to the point. And, even more, thanks for the suggestion, which we applied.

We modified the symbol for the true, unknown triggering window in section 4, which now is symbolized with a dagger instead of an asterisk. In addition, we edited the related text in section 5 as follows: "*Here we will provide an example in which the triggering time interval is assumed to be the time interval during which the most severe intensity was observed, meaning that we assume $W^\dagger=W^*$.*"

Furthermore, in the discussion of Figure 1b, we added: "**Here, the scaling of IW* pairs and ID pairs is similar because our definition of $W^*$ follows what is done in IDF curves, but the intensities associated with the true $W^\dagger$ may scale differently from the IDF curves, as they may be conceptually different.**"

We think the clarity is now much improved thanks to this suggestion.

Some readers may also wonder why the authors did not apply the same framework to a broader dataset that includes landslides or other regions. However, I recognize that the authors have used a well-constrained, high-quality debris-flow dataset, and that extending the same analysis to other phenomena or regions would be extremely challenging due to the inherent difficulties in identifying a sufficient number of ID pairs and determining reliable W* values. Therefore, I consider this dataset and the associated analysis to be particularly valuable. That said, these practical and methodological challenges may not be readily apparent to some readers, so providing a brief clarification in the text could further help convey the value and uniqueness of this dataset. For landslides, it is generally impossible to predict where they will occur in advance or to identify the exact time of initiation. The recurrence interval of landslides in a given region typically spans several decades to centuries, making it difficult to obtain a sufficient number of ID pairs at the regional scale. Consequently, most ID thresholds for landslides have been derived at the national scale. Preparing a landslide dataset suitable for an analysis such as that presented in Section 5 is therefore extremely difficult.

For debris flows, regional ID thresholds are often derived from ID pairs based on observed occurrences within the same or nearby catchments. However, in many cases, "occurrence" refers to the arrival of debris flow at the observation point rather than the initiation of motion. As mentioned in Comment 1 and the references therein, this implies that the threshold inherently includes the processes of debris-flow development and runout, which depend not only on rainfall but also on sediment availability, distribution, and composition. Hence, the strict identification of W* is practically difficult.

Thank you for these considerations. The reviewer agrees with our choices in terms of dataset and methods. The only request is to better explain the importance of the dataset following the concerns of the previous reviewers. Because it appears both reviewers are now satisfied, we believe no action is required in this instance.

The paragraph in Section 4 (L78-87) already mentions, at least in part, that W* is practically indeterminable. As mentioned earlier, I suggest moving that paragraph to Section 5 and expanding on it there. My recommendation is to strengthen the explanation of the practical difficulties in determining W* and in obtaining numerous ID pairs from real-world data. The authors could clarify that the analysis in Section 5 essentially deals with another metric (such as W corresponding to max(Tw)) but that, in this study area and dataset, assuming W = W* is reasonably valid, citing adequate references to support this rationale. This approach would clearly separate Sections 1–4 as describing theoretical principles and Section 5 as demonstrating an empirical application and interpretation based on real data.

Thank you for the suggestion. As mentioned above, we modified the symbol for the true, unknown triggering window in section 4, which now is symbolized with a dagger instead of an asterisk. In addition, we edited the related text in section 5 as follows: "*Here we will provide an example in which the triggering time interval is assumed to be the time interval during which the most severe intensity was observed*, **meaning that we assume $W^\dagger = W^*$.**"

Furthermore, in the discussion of Figure 1b, we added: ***"Here, the scaling of IW* pairs and ID pairs is similar because our definition of $W^*$ follows what is done in IDF curves, but the intensities associated with the true $W^\dagger$ may scale differently from the IDF curves, as they may be conceptually different.***"

I also agree with the discussion in L123-134. However, as noted in Comment 1, it is important to emphasize that the key finding, that W* ranges from 30 minutes to 6 hours, was derived specifically for debris flows. Whether landslides exhibit a similar pattern remains unknown. If the debris flows analyzed here are sourced primarily from channel-bed sediment, this relatively broad time interval may reflect temporal variations in sediment availability within the catchment (e.g., Tsunetaka et al., 2021a).

It is true that these results pertain to debris flows but, as mentioned above, the qualitative results hold irrespective of the processes. This simple real-world example is intended to make the theory more tangible. As mentioned in our responses to the previous review round, we

don't think we should dive into discussions about the triggering processes as this is way out of our scope. Actually, we probably should not, because we don't have sufficient elements for that type of reasoning.

3.                  Scaling                limitation                of                IDF                curves
The authors, for convenience, have estimated the return periods of very short-duration rainfall (less than 15 minutes) based on existing IDF curves. I am concerned that the validated lower limit of the existing IDF scaling (Borga et al., 2005) may be around 15-minute rainfall durations. The estimation of return periods for such short-duration rainfall involves high uncertainty. In fact, in Figure 1b, there appear to be at least two unrealistic data points plotted at less than 1 hour on the x-axis and around 200 mm h⁻¹ on the y-axis. Although I understand that there is currently no practical alternative approach, a brief mention of this limitation in the main text would further improve the clarity of the manuscript. I also believe that this concern is independently addressed by the results presented in Figure 3, which show the decorrelation time of rainfall. I was quite impressed by how closely this figure conceptually aligns with Figure 2. If space permits, adding a more detailed explanation and discussion of Figure 3 would make the manuscript even more refined and insightful.

This is a great point to address. The "unrealistic" values you mention concern an event that occurred over the night between July 16 and 17, 2007 in Passeier Valley. The reason we are familiar with this event is that the radar showed extremely large values, exceeding 100 mm over less than one hour, with no rain gauge data provided by the regional authorities to support such numbers. Certainly an "unrealistic" value, especially coming from the radar. However, radar maps suggested that the St. Leonard in Passeier station was directly hit by the event - a rare occurrence (Lengfeld et al., 2020). Further investigations revealed that the St. Leonard in Passeier data was removed during quality control due to "unrealistic" high values. Once collected, the "unrealistic" values from the rain gauge fully confirmed the "unrealistic" values from the radar. These numbers are real.

The work by Lengfeld et al. (2020) shows that the German rain gauge network, one of the densest in the world, captures less than 20% of the hourly extreme events. Together with the systematic sampling bias related to the analysis of landslide and debris flow triggering discussed in Marra et al. (2016) this explains why some very real values may appear as "unrealistic".

Lengfeld et al., 2020, ERL, https://doi.org/10.1088/1748-9326/ab98b4

Marra et al., 2016, JoH, http://doi.org/10.1016/j.jhydrol.2015.10.010

4.        Comparison        of        slopes        regarding        ID,        IW*,        and        IDF
The discussion comparing the slopes of ID, IW*, and IDF relationships may need to be moderated, as the current analysis does not provide sufficient evidence to draw a definitive conclusion. Because the D values of the ID pairs are relatively large, the data points in this dataset appear only within the range of approximately 1 to 48 hours on the x-axis in Figure 1b. The scaling for durations shorter than 1 hour is essentially extrapolated.

*Thank you for this suggestion, we amended the text to make it softer, as follows: "These latter threshold **seem to better** align with the regional scaling of extreme rainfall (dashed-dotted lines in the background), **suggesting that** the apparent difference in the power-law scaling of ID thresholds and IDF curves discussed by \citet{Bogaard2018} can be attributed to methodological issues in the choice of rain duration, often made regardless of the physical processes responsible for debris flow or landslide occurrence."*

Considering this, when focusing on the range between 1 and 48 hours in Figure 1b, the differences in slope among the ID threshold, IW* threshold, and IDF scaling appear to be nearly equivalent. Therefore, the overall difference in slope might simply reflect the data limitation that there are few ID pairs with small D values. For landslides, triggering rainfall events with D < 1 hour are extremely rare. However, for debris flows, such short-duration triggering events have been reported in various catchments (e.g., Abancó et al., 2016; Bel et al., 2017; Tsunetaka et al., 2021a).

*Thank you for raising this consideration, on which we fully agree. We added a sentence to this section to better highlight this aspect: "**Here, the scaling of IW\* pairs and ID pairs is similar because our definition of $W^*$ follows what is done in IDF curves, but the intensities associated with the true $W^\dagger$ may scale differently from the IDF curves, as they may be conceptually different."***

Line                                     by                                     line                                     comments
Title: Since the case study focuses specifically on debris flows, it might be helpful to include the term "debris flow" in the title to clearly indicate the study target.

*We respectfully disagree with this suggestion. The study uses debris flows as an example but it could have used synthetic data instead because it ultimately focuses on a conceptual problem that is not necessarily related to the triggering processes.*

L27-29: Readers who are less familiar with rainfall thresholds may wonder why the parameter E is sometimes used. A brief explanation of its meaning and rationale, supported by an appropriate reference, would help clarify this point.

*Thank you, we specified: "the total precipitation depth (for which the symbol E is typically adopted)"*

L50: IDF are -> IDF curves are

*Corrected, thank you*

L61: an user defined -> a user defined

Corrected, thank you

L78-79: Please consider softening the tone of the explanation slightly to make it more balanced and accessible to a broader readership.

Thank you for the suggestion, we rephrased the sentence to: *"It is therefore erroneous to quantify the **probability** of a **triggering event of** intensity $I$ in the ID space of the ID thresholds using **unconditional** probabilities from the IW space of the IDF curves"*.

L89-90: It would be helpful to clarify whether these debris flows were initiated by landslides or if they mainly resulted from bulking and entrainment of unconsolidated channel-bed material. A brief explanation would improve the reader's understanding.

Thanks for the suggestion. We included this information in the text, as follows: *"**We discuss the potential effects of the inconsistency highlighted above using data from 12 storms that triggered 133 debris flows in the eastern Italian Alps during 2005-2014. They constitute $\sim 40\%$ of all the debris flows recorded in the area in this period \citep{Nikolopoulos2014} and resulted from bulking and entrainment of unconsolidated channel-bed material.**"*

L98: It is not entirely clear whether these represent the triggering locations or the observation locations. Please clarify how the triggering location was defined in this dataset.

The mentioned sentence clearly states that these are the locations of the debris flow triggering: *"For each debris flow, we extracted the precipitation time series observed by the radar over the triggering locations."* To avoid possible confusion on this, we included additional information about the spatial accuracy of the database we used, as follows: *"**The triggering location of the debris flows in the database is provided with an uncertainty which is much smaller than the pixel size of the radar data (approximately one order of magnitude smaller, Marra et al., 2014)**."*

Figure 1: To further aid interpretation, you might consider adding summary scatter plots for all 133 events alongside Figure 1a: specifically, $W_*$ vs. $D$ and $I_*$ vs. $I$. Including such plots could make the relationships easier to grasp, particularly for readers who are less familiar with rainfall threshold analyses.

Thank you for this suggestion. We prefer to keep the figure as is to avoid overemphasizing the study case and distract the readers from the main message.

References
Abancó, C., Hürlimann, M., Moya, J., & Berenguer, M. (2016). Critical rainfall conditions for the initiation of torrential flows. Results from the Rebaixader catchment (Central Pyrenees). Journal of hydrology, 541, 218-229.
Bel, C., Liébault, F., Navratil, O., Eckert, N., Bellot, H., Fontaine, F., & Laigle, D. (2017).

Rainfall control of debris-flow triggering in the Réal Torrent, Southern French Prealps. Geomorphology, 291, 17-32.

Guo, X., Cui, P., Li, Y., Fan, J., Yan, Y., & Ge, Y. (2016). Temporal differentiation of rainfall thresholds for debris flows in Wenchuan earthquake-affected areas. Environmental Earth Sciences, 75(2), 109.

Pastorello, R., Hürlimann, M., & D'Agostino, V. (2018). Correlation between the rainfall, sediment recharge, and triggering of torrential flows in the Rebaixader catchment (Pyrenees, Spain). Landslides, 15(10), 1921-1934.

Tsunetaka, H., Hotta, N., Imaizumi, F., Hayakawa, Y. S., & Masui, T. (2021a). Variation in rainfall patterns triggering debris flow in the initiation zone of the Ichino-sawa torrent, Ohya landslide, Japan. Geomorphology, 375, 107529.

Tsunetaka, H., Shinohara, Y., Hotta, N., Gomez, C., & Sakai, Y. (2021b). Multi-decadal changes in the relationships between rainfall characteristics and debris-flow occurrences in response to gully evolution after the 1990–1995 Mount Unzen eruptions. Earth Surface Processes and Landforms, 46(11), 2141-2162.